# Aluminum Nitride Out-of-Plane Piezoelectric MEMS Actuators

**DOI:** 10.3390/mi14030700

**Published:** 2023-03-22

**Authors:** Almur A. S. Rabih, Mohammad Kazemi, Michaël Ménard, Frederic Nabki

**Affiliations:** Department of Electrical Engineering, École de Technologie Supérieure, Montréal, QC H3C 1K3, Canada

**Keywords:** out-of-plane actuation, piezoelectric actuator, MEMS, residual stress, aluminum nitride

## Abstract

Integrating microelectromechanical systems (MEMS) actuators with low-loss suspended silicon nitride waveguides enables the precise alignment of these waveguides to other photonic integrated circuits (PICs). This requires both in-plane and out-of-plane actuators to ensure high-precision optical alignment. However, most current out-of-plane electrostatic actuators are bulky, while electrothermal actuators consume high power. Thus, piezoelectric actuators, thanks to their moderate actuation voltages and low power consumption, could be used as alternatives. Furthermore, piezoelectric actuators can provide displacements in two opposite directions. This study presents a novel aluminum nitride-based out-of-plane piezoelectric MEMS actuator equipped with a capacitive sensing mechanism to track its displacement. This actuator could be integrated within PICs to align different chips. Prototypes of the device were tested over the range of ±60 V, where they provided upward and downward displacements, and achieved a total average out-of-plane displacement of 1.30 ± 0.04 μm. Capacitance measurement showed a linear relation with the displacement, where at −60 V, the average change in capacitance was found to be −13.10 ± 0.89 fF, whereas at 60 V the change was 11.09 ± 0.73 fF. This study also investigates the effect of the residual stress caused by the top metal electrode, on the linearity of the displacement–voltage relation. The simulation predicts that the prototype could be modified to accommodate waveguide routing above it without affecting its performance, and it could also incorporate in-plane lateral actuators.

## 1. Introduction

Photonic integrated circuits (PICs), and in particular silicon photonics, have developed tremendously over the past decade, because of the ever-increasing demand for fast and high-capacity optical communications. Nevertheless, further improvements in complexity, performance, and cost are impeded by the fact that different optical processing functions, such as light emission, modulation, filtering, switching, and detection, are best performed by devices fabricated with different materials. For instance, advanced lasers [1,2,3] and fast photodiodes [4,5] at telecommunication wavelengths are made with III–V materials. On the other hand, recent work on integrated lithium niobate modulators indicates that they can achieve record performance [6]. Silicon photonics, including devices made with silicon nitride waveguides, can be used to implement compact, low-loss passive [7,8,9,10] and tunable filters [11], and can be monolithically integrated with electronic components [12]. Integrating components made on different chips enables devices with record-breaking performance, as recently demonstrated with the narrow linewidth laser presented in [13], for instance. Nevertheless, misalignments between photonic and optical components are an important source of optical losses, especially when aligning active components such as laser sources and semiconductor optical amplifiers (SOAs). Despite the sub-micron accuracy of the pick-and-place tools implemented by the commonly used flip-chip bonding (FCB) technology, the issue of bonding alignment is still a major challenge [14], where out-of-plane and lateral in-plane misalignments were reported [15,16,17,18,19]. Beside the accuracy of pick-and-place tools, fabrication-dependent parameters were also found to contribute to total misalignments, where in the out-of-plane direction, maximum accuracies of only up to ±0.5 μm were achieved [18]. These misalignments are caused by the accuracy of the etching processes, the thickness tolerance of the deposited layers, and the bonding force [15]. MEMS actuators combined with suspended waveguides could be a solution to resolve these alignment issues by providing dynamic alignment at low cost and within compact structures. For instance, in [20], electrothermal bimorph actuators were proposed to compensate ~8 μm out-of-plane misalignment between the indium phosphide (InP) active chip and the silicon photonic chip.

Most used MEMS to provide out-of-plane motions include electrostatic [21,22], electrothermal [23,24], and piezoelectric [25] actuation mechanisms. The selection of the actuation mechanism depends on several factors, such as the availability of the fabrication technology, the maximum power dissipation allowed, and the maximum available voltage. Out-of-plane electrostatic actuation is performed using parallel-plate capacitors, staggered comb drives [26], asymmetric combs [27], or with vertical repulsive forces [28]. The staggered and asymmetric comb drive approaches need extra lithography, etch, and deposition steps during the fabrication process. For instance, multiple deep reactive ion etching (DRIE) processes are required [29], and in some cases the substrate is also patterned to create the lower comb drives [30]. Thus, they are more complex and costly [31]. On the other hand, repulsive forces require upper and lower electrodes that can also be complex to fabricate in silicon-on-insulator (SOI) processes that are favored for MEMS. As electrothermal actuators consume significant amounts of power, piezoelectric actuators are preferred for low-power applications.

Typically, piezoelectric actuation is not able to provide as large out-of-plane displacements [32] as that enabled by electrothermal actuations. Nonetheless, piezoelectric actuation provides fast response times at low voltage and with low power consumption [33]. In addition, the possibility of having both up and down precise out-of-plane displacements is expected to make piezoelectric actuation a good candidate for alignment between PICs. In [34], a piezoelectrically driven micro-lens was used to align an optical fiber to a photonic device. The lens was placed between the fiber and photonic device and was held from the top and bottom by the actuators to move it in x, y, and z axes to maintain the optical alignment. To make piezoelectric devices, ceramics made of lead zirconate titanate (PZT) have been widely used, due to their high piezoelectric properties [35]. For instance, the authors of [36] reported a large z-axis PZT actuator for an endoscopic microscopy application. The actuator provides as high as 120 μm of displacement at 20 V. In another study presented in [37], a PZT actuator with >3 μm of z-axis displacement using 170 V was proposed for high-force applications such as tactile displays and micropump applications. In [38], a dynamic PZT actuator was used to drive a micro-lens in the z-axis for miniaturized cameras, confocal microscopy, and pico-projectors. The actuator was controlled by a feedback system based on an optical displacement sensor. In another study [39], a combination of piezoelectric and electrostatic actuators was used to provide a 2D scanner to obtain vertical cross-sectional fluorescence images in an endomicroscope. A large commercial PZT-based z-axis actuator with an embedded strain gauge sensor was used to provide up to 400 μm vertical displacement, whereas an electrostatic actuator was used to provide a rotation on the second axis. As seen, PZT-based actuators, due to their high piezoelectric coefficients, provide large displacements at relatively low actuation voltages. However, PZT-based devices are difficult to use within integrated circuits [40]. In addition, recently, there have been environmental concerns about the use of lead in piezoelectric materials such as PZT [41]. Aluminum nitride (AlN) is one of the most studied non-ferroelectric piezoelectric thin films in the last decades, as its crystal is isotropic in the x-y plane but anisotropic along the z-axis [42]. AlN has been used as a piezoelectric thin film in many applications, including energy harvesters [43], microphones [44], inertial sensors [45], and bulk acoustic wave resonators [46]. Despite the relatively low piezoelectric coefficients of AlN, it remains well suited in applications such as chip-to-chip alignment, where small out-of-plane displacements are required, due to its unique characteristics, such as its high Young’s modulus, MEMS-CMOS compatibility [25], ease of deposition [47], and good optical properties [48], which make it a good candidate for integrated photonic devices. It is also lead-free, as opposed to other piezoelectric materials such as PZT, which is a significant benefit for commercial applications.

This work presents a novel out-of-plane piezoelectric actuator to control the vertical position of a suspended platform. The actuator includes a capacitive sensing mechanism to electrically monitor its motion. The novelty of this actuator is its ability to provide positive and negative out-of-plane displacements with a sensing capability, where the sensed capacitance is linearly dependent on the displacement of the platform. The structure is also uniquely designed to be able to support an optical waveguide. The purpose of this study is to pave the way for integrated positioning systems that can provide out-of-plane dynamic alignment between different types of PICs. One of the issues faced with the current active alignment technique used to assemble optical subsystems is that the misalignments occurring after assembling and bonding the subsystems remain significant sources of optical losses. Thus, the main outcome of developing the proposed alignment system based on the piezoelectric actuation of suspended waveguides will be to provide a low-cost, high-accuracy integrated mechanism to align components built on different chips. Moreover, such integrated structures can allow for alignment adjustments during operation and not only at the packaging step. This paper is organized as follows: Section 2 presents a schematic and the operating principle of the actuator along with the simulation method used to design the devices and the fabrication process utilized to build them. Section 3 describes the simulations and experimental results. In Section 4, the results are discussed, and finally in Section 5, conclusions are provided.

## 2. Materials and Methods

This section is divided into four parts. First, the operating principle of the actuator is explained, and a schematic is presented. Then, the steps followed to design and simulate the device are outlined. The third part describes the fabrication process followed, and finally, the procedures for testing the devices are explained in the fourth part.

### 2.1. Operating Principle and Schematic of the Actuator

The actuator is designed to be implemented on a SOI wafer, above which optical waveguides made of a silicon nitride core surrounded by a bottom and top cladding of silicon oxide could be patterned. As shown in the schematic presented in Figure 1, the function of the actuator is to precisely align the passive silicon nitride waveguide that lies on the platform with an optical active chip to optimize the optical coupling. Note that in our working assumptions, the active chip to couple to will be placed inside a cavity created on the SOI wafer. Thus, the range of motion of the actuator only needs to be large enough to compensate the variations in the etch depth of the cavity and in the thickness of the active chip, which are on the order of one micron or less. Since optical modes in integrated devices have dimensions of a few hundreds of nanometers, optical losses increase rapidly with misalignment, even at small values.

This actuator employs the piezoelectric effect to control the elevation of a platform that is used to emulate the waveguide carrier in a PIC. A schematic representation of the fabricated piezoelectric actuator is shown in Figure 1a, whereas Figure 1b shows a version of the device that is modified to accommodate the optical waveguides in future work. Simulated results of the modified version are provided in the Discussion Section. The actuator is used to support the platform and vertically align it to a fixed test structure. This test structure, which simulates the active chip, will be aligned with the suspended waveguide in this demonstration. The actuator consists of a central platform supported by two arms attached to two piezoelectric actuators. The two actuators are anchored to the substrate by folded beams. Each actuator consists of a 0.5 μm-thick aluminum nitride (AlN) layer as the active piezoelectric material, and a 10 μm-thick silicon-on-insulator (SOI) device layer with a resistivity in the range of 1–10 Ω-cm, that is used as the device layer and as the bottom electrode. The top electrode of the piezoelectric actuator is made of 1.0 μm-thick aluminum (Al) layer, which also serves as the metal to form contact pads for electrical connections. Two sets of capacitive combs, referred to by 1 in Figure 1, were added to track the out-of-plane motion in the upward direction, whereas combs denoted by 2 are meant for tracking the motion in the downward direction. Separate combs were used to monitor the upward and downward motions to simplify tracking the platform displacement in the two directions of motion, since the capacitance measurements do not indicate the direction of motion.

This actuator moves in response to mechanical strain imparted by an electric field across the AlN layer, arising from a potential difference between the two electrodes sandwiching the AlN (i.e., aluminum on top and SOI on the bottom). Longitudinal and transverse piezoelectric forces create the strain that initiates the motion [49]. As shown in Figure 1a, the two piezoelectric actuators are anchored by suspended folded beams and the H-shape platform is connected to these actuators through two long arms.

The direction of the platform displacement is controlled by the polarity of the applied electric field. The intrinsic electric field direction normally coincides with the positive z-axis; thus, applying a negative voltage on the top electrode generates an electric field that coincides with the intrinsic electric field, causing the upward motion. On the other hand, applying a positive voltage on the top electrode will reverse the operation, and thus the motion will be downward. The device was also designed in a way that provides the minimum stiffness in the out-of-plane direction, where the stiffness in the lateral directions is maximized by using a width to thickness ratio of 3, which is the minimum allowed by the fabrication process.

### 2.2. Design and Finite Element Analysis Simulations

The actuators were designed by taking into consideration the design rules of the commercial PiezoMUMPs process available through CMC. As the actuators are intended to be used for chip-to-chip optical alignment, the target displacement range was determined by the maximum out-of-plane misalignment reported with common chip integration techniques, such as flip-chip bonding, i.e., ±0.5 μm [18]. Therefore, the piezoelectric actuators shown in Figure 1 were designed to provide a displacement of 1 μm. The thicknesses of all the layers used in the PiezoMUMPs process are predefined and fixed; therefore, the displacement targets were achieved by simulating systematic variations of the lengths and widths of the actuators. Simulations showed that the optimal piezo-actuators must be 1.5 mm-long by 0.17 mm-wide. The required number and dimensions of sensing combs for a given sensing application was determined by multiple factors, including the desired sensitivity, sensing range, and fabrication constraints. Fundamentally, a higher number of comb fingers will provide a higher sensitivity and larger capacitance range. For the actuator, the number of sensing comb fingers was determined by the overall area of the piezoelectric actuator that was used to meet the 1 μm displacement target, as well as the minimum finger width and gaps allowed by the design rules (i.e., 3 μm). Therefore, the maximum number of fingers that could be used was 132 fingers for upward displacement sensing and 160 for downward displacement sensing.

The device was simulated numerically using the finite element analysis (FEA) software CoventorWare. Correct mesh sizing for the different designed parts and setting of the piezoelectric coefficients played a significant role in achieving accurate device modeling. The design and simulation of the device followed the rules of the PiezoMUMPs fabrication process provided by MEMSCAP [50], and the reverse piezoelectric effect mode was selected in the software to accurately model the behavior of the actuators. The piezoelectric strain was simulated with d31 and d33 piezoelectric coefficient values for AlN of −2.6 pm/V and 5.5 pm/V, respectively. The other coefficients and elastic constants used are available in [49]. It is worth mentioning that the signs of the piezoelectric coefficients were reversed from those reported in [49], in order to obtain the displacement in the same direction observed by the measurements.

### 2.3. Fabrication Process

The fabrication of this device used five masks, as per the PiezoMUMPs process [50]. As shown in Figure 2a, the process started with a 150 mm n-type, double-side-polished silicon-on-insulator (SOI) wafer, with a handle layer thickness of 400 ± 5 μm.

The top surface of the silicon was doped by phosphosilicate glass (PSG) and annealed at 1050 °C for 1 h in argon, and then the PSG was removed using chemical wet etching. Then, a 0.2 μm-thick oxide layer was thermally grown and patterned using the first mask (oxide mask) through reactive ion etch (RIE) to define the ground pad and the device area where the piezoelectric material is directly attached to the 10 μm-thick SOI device layer, as shown in Figure 2a,b. The second mask (Figure 2c) was used to pattern a 0.5 μm-thick aluminum nitride (AlN) piezoelectric layer by wet etching. The third mask (Figure 2d) was used to define a metal stack of 20 nm of chrome and 1 μm of aluminum for the pads and electrical routing, patterned using a liftoff process. With the fourth mask, the SOI layer was etched from the front side using deep RIE (DRIE), as shown in Figure 2e. Thereafter, a polyimide coat was used (Figure 2f) as a front-side protection material to cover the top surface of the SOI layer, and the wafers were then flipped and back-side-etched using the fifth mask. First, RIE was used to remove the bottom thermally grown oxide, then a deep trench within the substrate back-side was patterned using DRIE, as shown in Figure 2g. Finally, the devices were released (Figure 2h) by removing the buried oxide (BOX) layer using wet etching and stripping off the front-side protection polymer using a dry etching process. An optical microscope image of the fabricated devices is shown in Figure 3.

### 2.4. Procedures for Experimental Testing

Two types of tests were performed. First, the displacement of the platform and the capacitance change as a function of DC voltage applied to the piezoelectric actuator were measured. Then, frequency sweeps of the excitation were carried out and the mechanical response of the device was measured with a vibrometer to determine the resonant frequencies and their associated modes. The movable SOI layer of the device was used as the common ground for both actuation and capacitive sensing, whereas the top Al was used as an active piezoelectric electrode. Before characterizing the device, the die was bonded on a specially designed PCB.

#### 2.4.1. Static Characterization

The static displacement was characterized using a LEXT OLS 4100 laser confocal microscope from Olympus, Tokyo, Japan, whereas the capacitance was measured using an AD7747 capacitive readout circuit [51]. The LEXT OLS 4100 provides a lateral resolution of 0.12 μm and a 10 nm height resolution, which is useful for the characterization of the out-of-plane displacement. Before using the microscope to measure the displacement of the platform, the height differences in a multilayered test structure were measured for calibration. The test structure was designed beside the platform to represent the active chip shown in Figure 1b, and to facilitate the out-of-plane displacement measurements. The test structure contains the SOI device layer, oxide, and AlN and Al layers, as shown in Figure 4.

Table 1 shows the average measured height differences compared to their values provided in the design rules handbook of the PiezoMUMPs process [50]. The table also includes the height differences between the top of the device layer and the oxide, the device layer and the AlN, and the device layer and the Al.

Considering the fabrication tolerances [50], these values are in good agreement with the expected ones. Having tested the accuracy of the microscope, before actuating the device, the initial deformation of the platform without an actuation voltage applied was measured against a fixed reference point, as shown in Figure 5. The platform was found to be below the reference point by 5.32 μm. The initial height difference between the platform and the reference point was set as the reference value to measure the displacement of the platform.

Then, a voltage in the range of −60 V to 60 V in steps of 10 V was applied to the actuators. At each step, an image was acquired and the height difference between the platform and the reference point was recorded. To find the displacement at a specific voltage, the height difference between the platform and the reference point at that voltage was calculated.

#### 2.4.2. Procedure for the Characterization of the Frequency Response

A Polytec laser doppler vibrometer (OFV2570 controller and OFV-534 laser unit) was used to determine the resonant frequency modes of the device. To excite the resonant modes, a function generator was used to apply the AC voltage signal to the electrodes of the piezoelectric actuator, and the laser beam of the vibrometer was focused onto the platform, where the reflected signal was monitored to measure its motion. An AC signal with a frequency ranging from 6 kHz to 15 kHz was applied, and a fast Fourier transform (FFT) was performed on the vibrometer signal to identify the resonant out-of-plane modes.

The first three resonant modes of the actuator identified using modal harmonics simulations are shown in Figure 6. All modes correspond to motion in the out-of-plane direction, despite that in the second mode the platform becomes a node, as shown in Figure 6b. The first mode occurred at 7.785 kHz (Figure 6a), and the second and third modes occurred at 11.730 kHz (Figure 6b) and 12.970 kHz (Figure 6c), respectively.

## 3. Results

### 3.1. Static Characterization

#### 3.1.1. DC Displacement

It was observed that applying negative actuation voltages to the top electrode (active electrode) caused the platform to move in the upward direction along the z-axis, whereas applying positive voltages led to motion in the downward direction. The measurement of the displacement under DC voltage excitation was performed using a confocal microscope, following the method previously described. Figure 7 shows microscope images of three measurements: at rest (0 V), maximum negative voltage (−60 V), and maximum positive voltage (60 V).

In total, three devices of the same design as the one shown in Figure 1a were tested. The devices were fabricated during the same manufacturing run but are from different dies. Throughout the following analysis, the three devices will be referred to as D1, D2, and D3. The tests were repeated five times for each device, and the averages were compared with the FEA simulation performed with the CoventorWare software, as shown in Figure 8.

The standard deviation of the five measurement runs for each device showed small variations within ±34 nm, which demonstrates the repeatability of the measurements across the devices and dies. It is worth pointing out that the upward displacement was larger than the downward displacement at the same actuation voltage. Initial downward bending of the platform caused by tensile stress from the top metal electrode was found to be the main cause. This made the device bend downward at rest, and thus limited the linear downward displacement achieved though the positive actuation voltage. As a result, the downward displacement achieved beyond 40 V was not linear with applied voltage. For negative bias from −10 V to −60 V, the average of the positive displacement of the platform for the three devices went from 0.09 ± 0.03 μm (0.12 μm for simulations) to 0.82 ± 0.03 μm (0.87 μm for simulations). On the other hand, for the positive bias from 10 V to 60 V, the average of the negative displacement of the platform went from −0.08 ± 0.03 μm (−0.11 μm for simulations) to −0.36 ± 0.03 μm (−0.51 μm for simulations). The results and the effect of the stress will be discussed in the Discussion Section.

#### 3.1.2. DC Capacitance

The capacitance of the comb fingers denoted by 1 in Figure 1a was recorded as a function of the displacement in the upward direction, whereas for downward displacements, the change in the capacitance was monitored using comb fingers denoted by 2 in Figure 1a. The capacitance changes in a device measured with the AD7747 and the simulated values are shown in Figure 9, versus the actuation voltage. Note that the capacitance was measured on an unpackaged bare die of the same design for a single device. This was needed in order to accurately measure the comb drive capacitances using the read-out circuit. The device was tested with a probe station (EP6 from Cascade) to minimize the effect of the parasitic capacitances of the read-out circuit caused by the wire-bonds, package, and test PCB used to achieve the displacement measurements previously presented.

For the entire actuation range in the upward direction and for voltages below 40 V in the downward direction, the actuator had a linear behavior. Beyond 40 V in the downward direction, the saturation in the capacitance was due to the reduced displacement caused by the nonlinear behavior, as explained above. At −60 V, the average change in capacitance over five runs was found to be −13.10 ± 0.89 fF (−12.29 fF for simulation), whereas at 60 V the change was 11.09 ± 0.73 fF (15.92 fF for simulation). Since the purpose of measuring the capacitance is to track the displacement of the platform, the capacitance change of this device was compared to the corresponding average of the average out-of-plane displacements of D1, D2, and D3, where the capacitance was found to decrease by increasing the out-of-plane displacement of the platform. For the downward displacement, at the maximum platform displacement of −0.36 ± 0.03 µm, the average capacitance change over five runs was 11.10 ± 0.74 fF (15.92 fF in simulations), whereas in the upward direction, the average measured capacitance change at the maximum platform displacement of 0.82 ± 0.03 µm was −13.10 ± 0.89 fF (−12.29 fF in simulations).

### 3.2. Mechanical Frequency Response

Figure 10 shows the mechanical frequency response extracted from D1 by using the vibrometer-based method previously discussed. The vibrometer was able to capture the first and the third modes at 7.841 kHz and 13.586 kHz, respectively. As shown in the simulation results presented in Figure 6b, in the second mode at 11.730 kHz, the platform of the actuator is a node that exhibits no out-of-plane displacement discernable by the vibrometer. Thus, only the first and the third modes can be measured by the vibrometer. Table 2 shows the measured resonant frequencies of all three devices in comparison with simulations.

## 4. Discussion

Three devices from three different dies were tested and the results were compared to the FEA simulations carried out using the CoventorWare software. As shown in Figure 8, the results of all devices showed good agreement, especially for negative actuation voltages. However, D1 provided the closest results to the simulations compared to the other two devices. At −10 V, the simulations predicted a displacement of 0.12 µm, but the average measurement over five cycles for each of the three devices (i.e., D1, D2, and D3) yielded 0.09 ± 0.03 µm, 0.08 ± 0.02 µm, and 0.10 ± 0.02 µm, respectively, resulting in average differences of 22.3%, 30.9%, and 17.3%, respectively. At the maximum applied negative voltage of −60 V, a displacement of 0.87 µm was simulated versus the measurement of 0.84 ± 0.02 µm, 0.83 ± 0.02 µm, and 0.79 ± 0.04 µm, for D1, D2, and D3, respectively, with percentage errors of 3.0%, 4.5%, and 8.7%, respectively. For positive voltages, at 10 V, the simulation yielded a displacement of −0.11 µm, whereas the measurement was −0.09 ± 0.02 µm, −0.07 ± 0.02 µm, and −0.09 ± 0.04 µm for D1, D2, and D3, respectively, leading to differences of 11.0%, 32.7%, and 17.0%, respectively. At the maximum applied positive voltage of 60 V, the simulation yielded −0.51 µm versus −0.46 ± 0.02 µm, −0.31 ± 0.03 µm, and 0.32 ± 0.05 µm for the measurement for D1, D2, and D3, respectively, and hence the differences were 9.9%, 38.2%, and 36.3%, respectively. Compared to actuators made of materials with higher piezoelectric coefficient, e.g., PZT [52], the 60 V required to provide um-level actuation on AlN piezoelectric actuators is relatively high. Nevertheless, it remains below the breakdown voltage, which is estimated to be 200 V (breakdown voltage: 4 MV/cm) [53].

Residual stress was found to have a significant impact on displacement in simulations, and as such, it is understood to be the main cause that led D2 and D3 to show larger differences to simulations than D1. Experiments showed that while D1 had a −5.31 ± 0.01 µm platform initial deformation with respect to the reference point shown in Figure 5, D2 and D3 showed −7.75 ± 0.40 µm and −7.67 ± 0.05 µm, respectively. The simulated results were obtained at a residual tensile stress of 165 MPa on the metal layer, which resulted in an initial simulated deformation (−5.25 µm) of the platform at 0 V. This matches rather closely the value measured for D1 at rest. Positive displacements caused by negative actuation voltages were always higher than negative displacements caused by the positive actuation voltages. For instance, in Figure 8, simulations showed that the platform can move by 0.87 μm with a −60 V bias. However, when −60 V was applied to the actuator, the maximum displacement was only −0.51 μm. Two factors can explain this phenomenon: the residual tensile stress applied to the device and the variation in the thickness of the layers (mainly the electrodes) on top of the structure. The former is expected to have the most significant impact. The residual stress is caused by mismatch either in the thermal expansion coefficients of the different layers, or in the lattice constants of the layers [42]. To show the effect of variations in the residual stress on the performance of the device, different levels of both compressive (C) and tensile (T) stresses were simulated by applying the stress on the metal layer that was used as the top electrode. The effect of stress was studied over the same actuation voltage range (±60 V) used experimentally. The correlation factor of the displacement–actuation voltage relation was extracted for each stress level. Simulations showed that the nonlinearity between the displacement and the actuation voltage increased with an increase in the stress level, as shown in Figure 11.

Stress levels in a range of ±265 MPa were investigated. For compressive stresses, the linearity coefficients were found to be ~1, 0.98, and 0.98 for stress levels of 65 MPa, 165 MPa, and 265 MPa, respectively, whereas for tensile stresses, the linearity coefficients were found to be 0.97, 0.95, and 0.90 for stress levels of 65 MPa, 165 MPa, and 265 MPa, respectively. On the other hand, the model without stress in any materials showed a linearity coefficient of 0.99. Tensile stress had a stronger effect on the linearity of the relation between the displacement and the actuation voltage than compressive stress. For the case of tensile stress, the displacement for positive actuation voltages was lower than the one obtained with negative actuation voltages, as proven by the experiments for all the tested devices. This is explained by the fact that under tensile stress, the beams that support the platform bend downward. Simulations predicted this fact, and this was confirmed during testing by the fact that the platform was below the level of the reference point in Figure 5. Positive actuation voltages caused the beam to move further downward, as explained in Section 2.1. Thus, the beam may already be approaching the limit of its elasticity. On the other hand, negative actuation voltages caused the beam to move in the opposite direction to the one caused by the residual stress. Hence, the displacement was higher, as demonstrated by both simulation and experimental results.

It is worth mentioning that the piezoelectric layer could also be hosting stress. A compressive residual stress of ~−46.3 ± 0.66 MPa was measured in aluminum nitride films in [42]. However, depending on the film deposition conditions, such as the temperature and pressure, the residual stress may vary, and can be as high as 600 MPa [54]. Besides stress, other factors could also contribute to the variations in results. Such factors include the accuracy of the measurement tools, the fabrication tolerances, and possible variations in the piezoelectric coefficients used, in addition to variations in material properties such as density, Young’s modulus, and the coefficient of thermal expansion. The relation between the capacitance of the sensing combs and the actuation voltage was found to be linear. This could enable a simple feedback control circuit to track the motion of the platform.

As presented, the aim of the fabricated devices is for their use as a waveguide positioner for active chip alignment. Thus, the modified version that has an optical path included for waveguide routing shown in Figure 1b was simulated and the results are compared in Figure 12 to the fabricated version previously discussed. The modified version has less piezo-material and metal area deposited on top of it, to make room for routing the waveguide.

Simulations indicate that the modified version of the device under a tensile stress of 100 MPa in the metal layer will provide the same performance as that of D1 at 165 MPa. Thus, the net residual stress of the modified version is expected to be reduced. Nonetheless, the waveguide is expected to also have an impact on the overall stress on the device and this will be characterized in future work.

As in-plane displacements are also required for the efficient waveguide positioning and chip-to-chip alignment, the width of the horizontal connecting arms could be reduced to integrate in-plane actuators in the future work.

Comparing our devices with the published works, despite the extensive use of piezoelectric transducers in several applications, including energy harvesters [43], microphones [55], inertial sensors [56], and bulk acoustic resonators [57], to the best of our knowledge, there are no works focusing on the PIC waveguide alignment application. The only report found for PICs’ alignment was presented in [34] to align an optical fiber with a photonic device. In that work, a ball lens controlled by piezoelectric actuators was placed between the fiber and the photonic device to maintain the optical alignment. Only a few demonstrations of piezoelectric-actuated optical systems were surveyed in the literature, but they were built for different applications. For instance, the authors of [58] used a AlN nano-bender piezoelectric actuator in a photonic crystal cavity for optical resonance tuning on nanometer scales. In [59], an acousto-optical modulator controlled through an AlN actuator was demonstrated. Work in [60] demonstrated phase and amplitude modulators in silicon nitride photonics controlled with an AlN actuator. Table 3 summarizes some of the works published on MEMS piezoelectric actuators used for different applications. As shown in the table, majority of the reported applications require displacements in the nanometer scale. Thus, actuators having sizes of hundreds of nanometers to a few micrometers were reported. The actuators in our proposed waveguide positioners for chip-to-chip alignment require the use of larger structures to support suspended waveguides and allow for a sufficient range of out-of-plane motion.

## 5. Conclusions

This work presented a novel out-of-plane aluminum nitride piezoelectric MEMS actuator. Testing the fabricated device at voltages ranging from −60 to 60 V yielded a maximum travel range of 1.30 ± 0.04 µm. The device is equipped with a capacitive sensing mechanism to track its displacement, where the measured capacitance has shown a linear relation with the displacement. Frequency sweeps of the input signal yielded mechanical resonance frequency modes close to the simulated values performed by FEA modeling. The effect of residual stress was also investigated, where measurements and FEA simulations were matched to quantify the stress on the fabricated devices. Tensile stress influenced the performance of the devices and caused the displacement to be nonlinear with respect to the actuation voltage. The initial investigation showed that larger dimensions (length and width) are required to increase displacement, however, this will increase the impact of residual stress, and hence increase the bending of the device that causes the nonlinearity. Thus, future work will investigate the effect of the geometry of the device on its response. In addition, in-plane actuators and waveguides on top of the suspended platform will be integrated.

## Figures and Tables

**Figure 1 micromachines-14-00700-f001:**
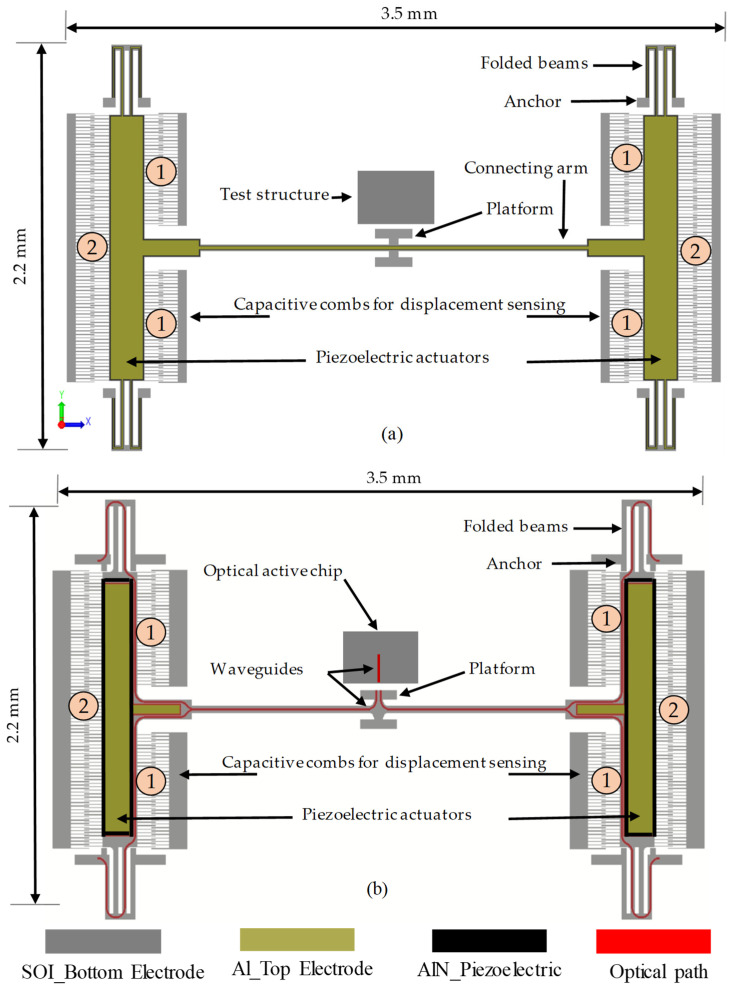
Schematic of the out-of-plane piezoelectric actuator: (**a**) the fabricated device, where the H-shape platform is aligned to a fixed test structure, and (**b**) a modified version (not fabricated) that accommodates waveguides on the platform for optical routing to align to an active optical chip.

**Figure 2 micromachines-14-00700-f002:**
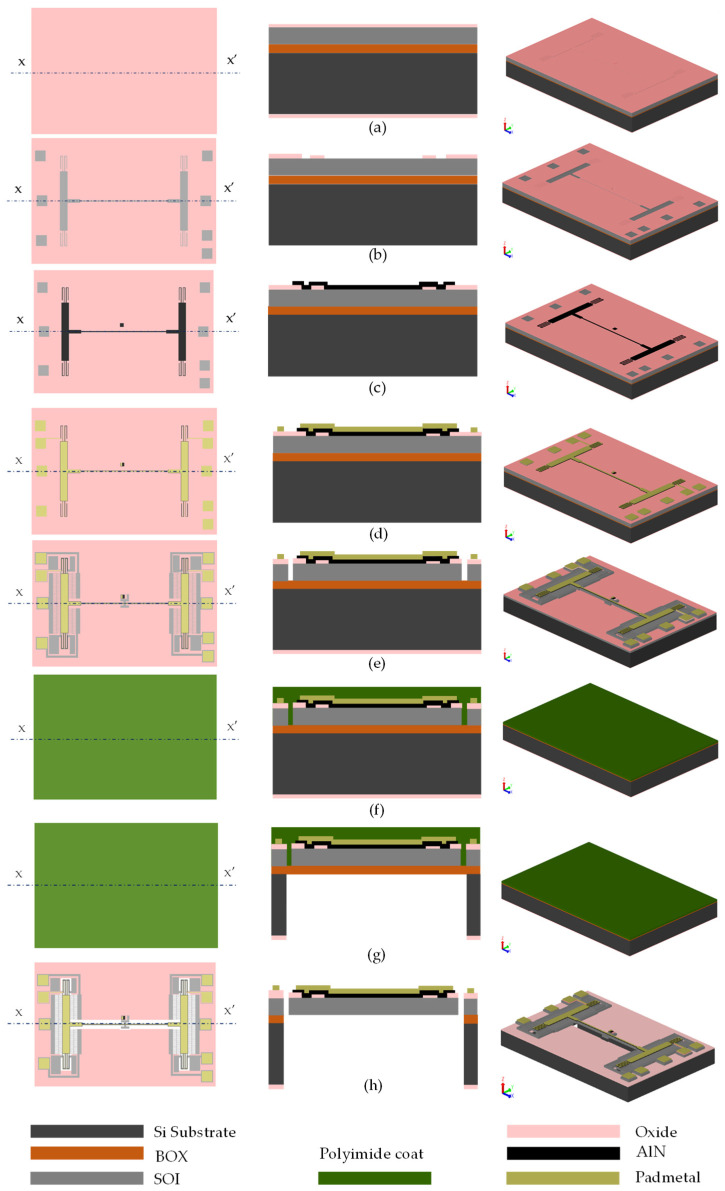
Fabrication steps of the piezoelectric actuator based on the PiezoMUMPs process: (**a**) the starting SOI wafer, (**b**) patterning the thermally grown oxide layer, (**c**) depositing and patterning of the AlN piezoelectric layer, (**d**) depositing and patterning the metal layer, (**e**) front-side etching of the SOI layer, (**f**), front-side protection layer using a polymer layer, (**g**) back-side etching of handle layer, and (**h**) releasing the device.

**Figure 3 micromachines-14-00700-f003:**
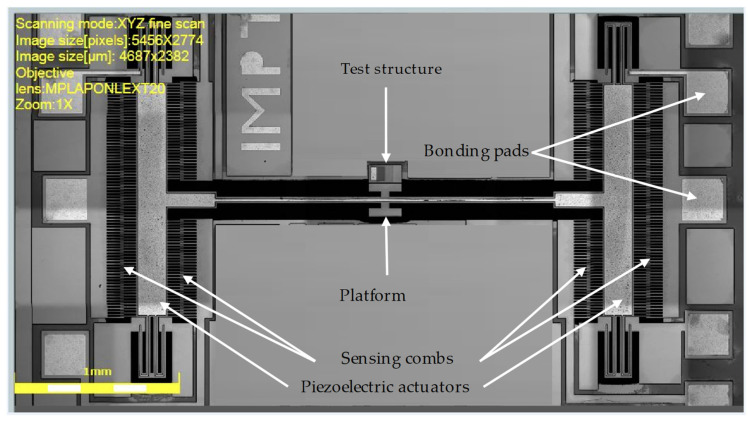
Optical micrograph of the fabricated MEMS device.

**Figure 4 micromachines-14-00700-f004:**
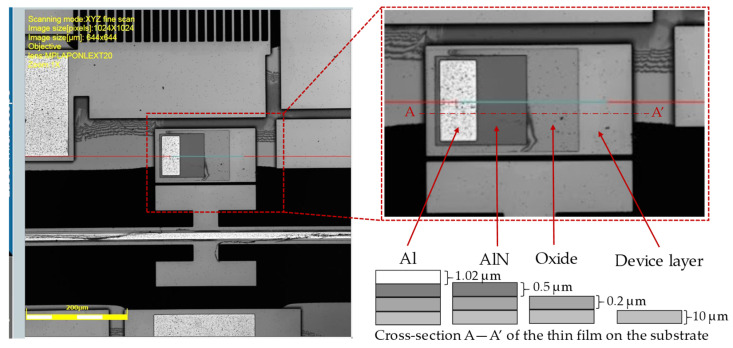
Multilayered test structure used for differential height measurements.

**Figure 5 micromachines-14-00700-f005:**
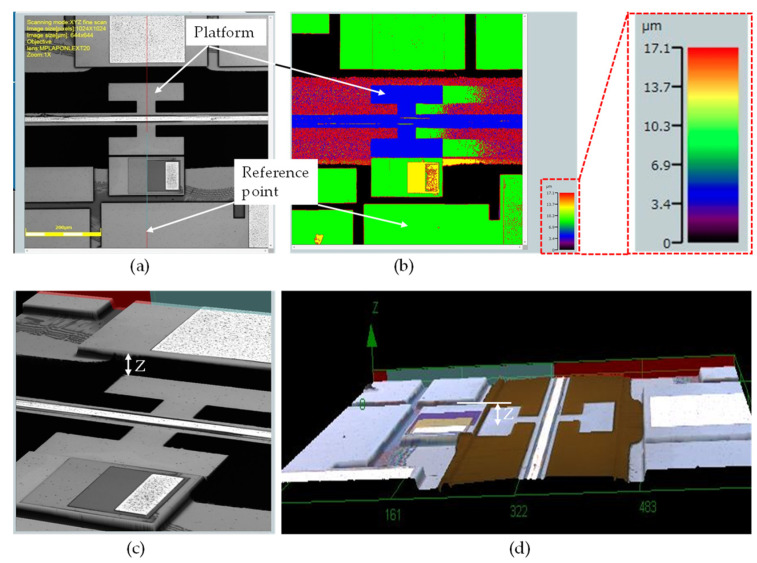
Initial deformation of the platform below the reference point due to residual stress: (**a**) the platform and reference measurement point, (**b**) color image highlighting the different levels of the platform and reference point, and (**c**,**d**) 3D pictures showing the z-axis height between the platform and the reference level.

**Figure 6 micromachines-14-00700-f006:**
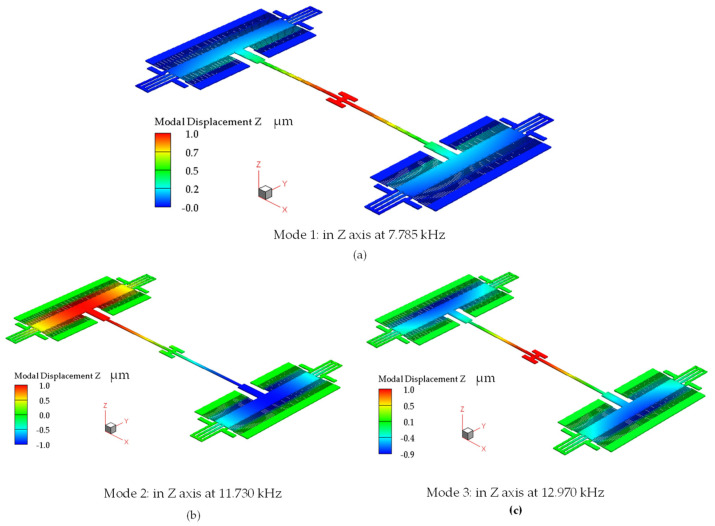
Simulated resonance frequencies showing (**a**) mode 1, (**b**) mode 2 (note: platform becomes a node), and (**c**) mode 3.

**Figure 7 micromachines-14-00700-f007:**
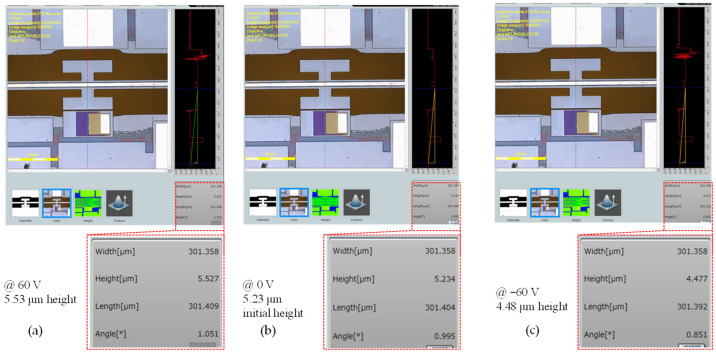
Microscope images showing the height of the platform with respect to the reference point for (**a**) −60 V, (**b**) at rest, and (**c**) at 60 V.

**Figure 8 micromachines-14-00700-f008:**
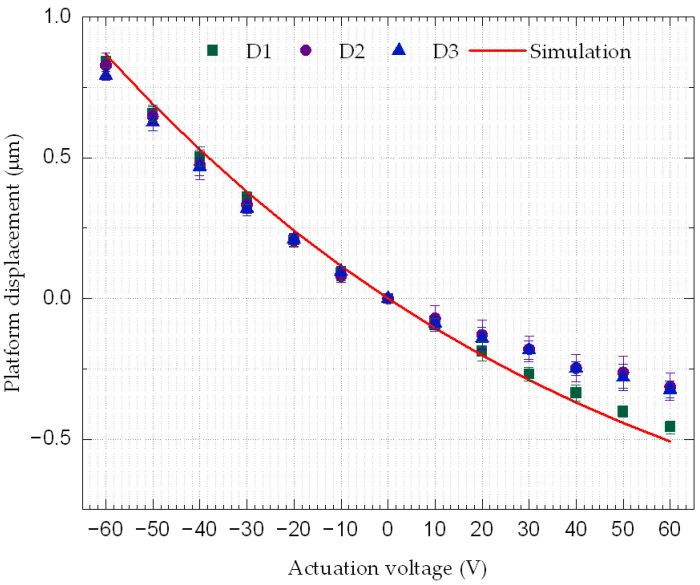
Averaged displacement for five measurement runs of three devices (D1, D2, and D3) versus the actuation voltage (the line represents the predicted behavior from the FEA simulation).

**Figure 9 micromachines-14-00700-f009:**
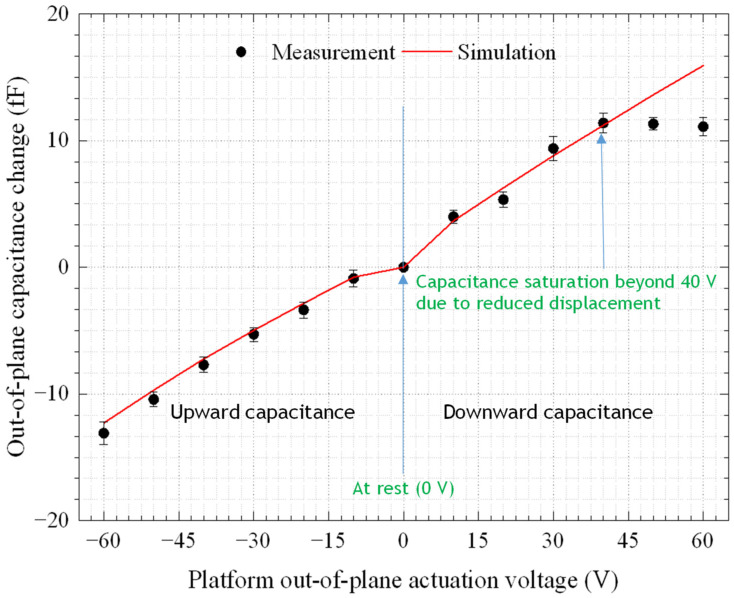
The average DC capacitance change versus the actuation voltage measured over five runs and the simulation results.

**Figure 10 micromachines-14-00700-f010:**
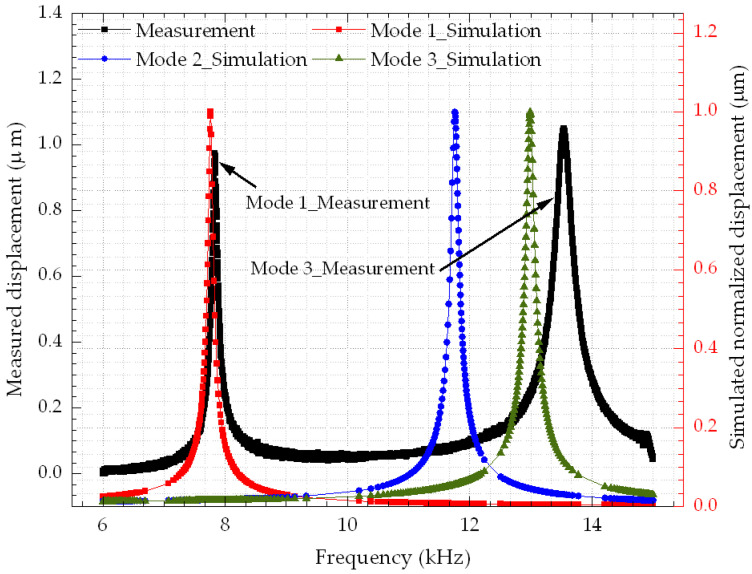
Measured frequency response of D1 showing two resonant modes along with the simulation results.

**Figure 11 micromachines-14-00700-f011:**
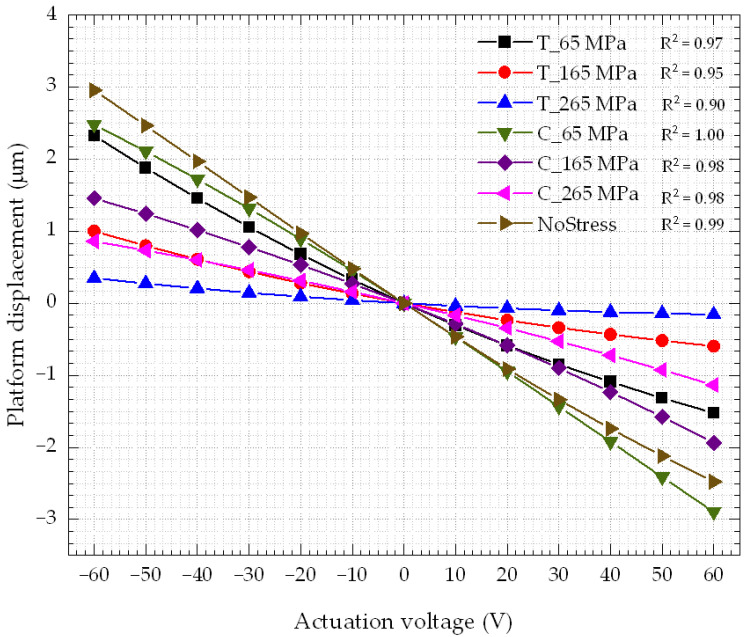
Effect of the stress on the linearity of the response of the actuator.

**Figure 12 micromachines-14-00700-f012:**
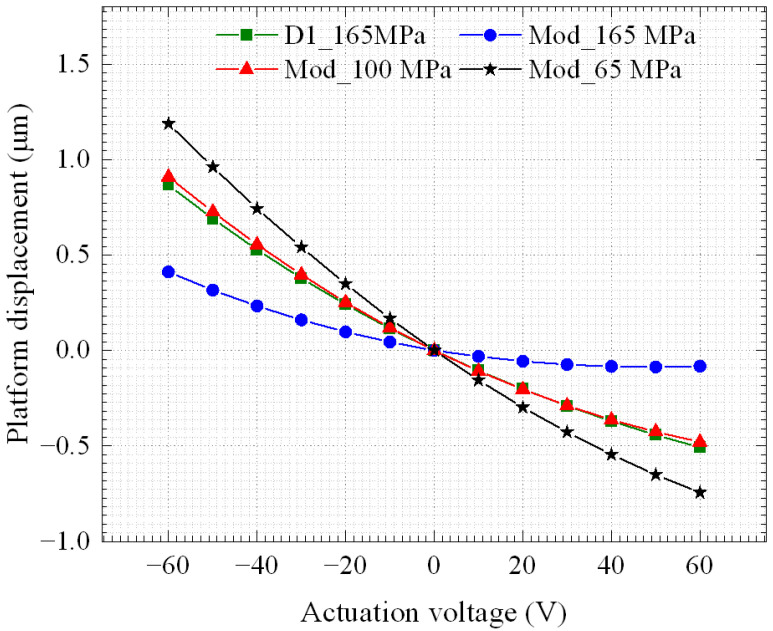
Out-of-plane displacement of the platform of the modified (Mod) version, including optical waveguides with different residual stresses compared to the fabricated devices (D1).

**Table 1 micromachines-14-00700-t001:** Measured height differences versus designed values.

Layers	Height Difference (μm)	
	Design Rules Value	Measured Value
Device layer to oxide	0.2	0.16 ± 0.06
Device layer to piezoelectric layer	0.7	0.58 ± 0.06
Device layer to metal	1.72	1.54 ± 0.06

**Table 2 micromachines-14-00700-t002:** Measured resonant frequency modes of the devices.

Die	Resonant Frequency (kHz)		Error%	
	Mode 1	Mode 3	Mode 1	Mode 3
D1	7.841	13.586	0.714	4.534
D2	7.992	13.772	2.590	5.823
D3	7.722	13.492	0.816	3.869
Simulation	7.785	12.970		

**Table 3 micromachines-14-00700-t003:** Performance of published MEMS actuators for waveguide alignment.

Application	Actuation Voltage (V)	Displacement (μm)	Ref.
RF switch	25	0.65	[54]
Logic applications	6	0.12	[61]
General use	120	0.21	[62]
Low-power logic	6	0.07	[63]
Phase shifting	2	1.25 × 10^−4^	[60]
Tuning optical resonance	3	0.18	[58]
Alignment	60	1.30	This work

## Data Availability

The data are available upon justifiable request.

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
