# Peer review of "Aluminum Nitride Out-of-Plane Piezoelectric MEMS Actuators"

_micromachines, 2023, doi:10.3390/mi14030700_

Round 1
Reviewer 1 Report
In this manuscript, the authors presented a AlN based piezoelectric MEMS actuator for PICs align different chips. Capactiance displacement sensor was intergrated for the sensing. It was claimed that the prototypes can output 1.3um displacement with ±60 V excitaiton. This idea can prototype is interesting. Adn this manuscript was well written.
Comments:
1. AlN has very low piezoelectric coefficients. Would the other piezo materials benefit the device?
2. 60V is a large one for MEMS device. How to apply such a large voltage in a small device?
3. Would the capacitance sensor have nonliearity?
4. Why a small hollow canbe observed in Fig. 7? which breaking the linearity
Author Response
The authors would like to thank the reviewer for the valuable comments. Please find our response attached.

Reviewer 2 Report
This paper proposed that out-of-plane piezoelectric MEMS actuators with a capacity detection of displacement. The proposed actuators has verified performance well experimentally, but this paper, to be published, should be provided with additional matters as following.
1. The authors claim that the actuator proposed is new but there are several existing studies of out-of-plane MEMS actuators based on PZT with capacity sensing availability. For example, Zhen Qiu and etc published “Targeted vertical cross-sectional imaging with near-infrared dual axes confocal fluorescence endomicroscope”. In this paper, they proposed 2-d PZT actuators (vertical displacement and rotation) even with capacitive sensing. So, the authors should provide differences and novelty with respect to this paper.
2. Further the point 1, the authors should provide comparison to other existing studies with similar structure (PZT, out-of-plane motion, capacitive sensing).
3. Although the authors claim that they proposed an actuator structure, they don’t explain how to design the actuator. They should provide design process such that (1) how to decide the shape and size of the PZT actuator, (2) relation to actuation force and displacement, (3) how to decide the number of sensing combs, (4) relation to displacement sensing capability, and so on.
4. Because this paper is discussed about MEMS system, the authors should show MEMS fabrication process step by step. Of course, they mentioned they followed MUMPs and MEMSCAP. However, certainly there will be their own fabrication parameters and process.
5. Figure 4 is difficult to understand. Height difference cannot be seen well.
6. Figure 6 shows the measurement results by checking DC output voltage. need to provide pictures taking from microscope and comparision to each other
7. Figure 6 shows measurement results of D1, D2, and D3 case. Because the experiment cases are three, simulation results also should show each case and compare to each other.
8. Figure 7 needs to show D1, D2, D3 case too.
Author Response

(The authors gave the same response as above.)

Round 2
Reviewer 2 Report
The author have completed all my review comments. Hence, the paper can be published.